# The Frequency and Healthfulness of Food and Beverage Advertising in Movie Theatres: A Pilot Study Conducted in the United States and Canada

**DOI:** 10.3390/nu12051253

**Published:** 2020-04-28

**Authors:** Stanley Wong, Elise Pauzé, Farah Hatoum, Monique Potvin Kent

**Affiliations:** 1School of Medicine, Faculty of Medicine, University of Ottawa, 451 Smyth Road, Ottawa, ON K1H 8M5, Canada; swong021@uottawa.ca; 2School of Epidemiology and Public Health, Faculty of Medicine, University of Ottawa, 600 Peter Morand, Room 301J, Ottawa, ON K1G5Z3, Canada; epauz022@uottawa.ca; 3Milken Institute School of Public Health, George Washington University, 950 New Hampshire Ave, Washington, DC 20052, USA; fhatoum@uottawa.ca

**Keywords:** obesity, children, food marketing, food environment, movie theatres, United States, Canada, policy, self-regulation, alcohol

## Abstract

The marketing of unhealthy foods and beverages contributes to childhood obesity. In Canada and the United States, these promotions are self-regulated by industry. However, these regulations do not apply to movie theatres, which are frequently visited by children. This pilot study examined the frequency and healthfulness of food advertising in movie theatres in the United States and Canada. A convenience sample of seven movie theatres in both Virginia (US) and Ontario (Canada) were visited once per month for a four-month period. Each month, ads in the movie theatre environment and before the screening of children’s movies were assessed. Food ads were categorized as permissible or not permissible for marketing to children using the World Health Organization’s European Nutrient Profile Model. There were 1999 food ads in the movie theatre environment in Ontario and 43 food ads identified in the movie theatre environment in Virginia. On average, 8.6 (SD = 3.3) and 2.2 (SD = 0.9) food ads were displayed before children’s movies in Ontario and Virginia, respectively. Most or all (97–100%) food ads identified in Virginia and Ontario were considered not permissible for marketing to children. The results suggest that movie theatre environments should be considered for inclusion in statutory food marketing restrictions in order to protect children’s health.

## 1. Introduction

As of 2016, 340 million children around the world have excess weight or obesity [1]. In the United States, nearly one in five children aged 2–19 years (18.5%) have obesity and 13.1% of children aged 6–17 years in Canada are in the same weight class [2,3]. Considered a major risk factor for chronic illnesses such as Type 2 diabetes, insulin resistance, and hypertension, obesity is a major public health concern [4]. It decreases both child and adult quality of life and the cost of treating the preventable diseases associated with obesity is a burden on healthcare systems around the world [4,5,6,7].

One factor contributing to obesity and poor dietary behaviours amongst children is the pervasive presence of unhealthy food and beverage marketing in traditional media (television and print media), new media (digital and social media), and in schools [8,9,10,11,12]. The marketing of these foods and beverages affect the preferences, food requests, and food intake of children [9,13,14,15]. Most food and beverages marketed to children have been deemed unhealthy as they contain high levels of sugar, fat, sodium, and calories [16,17]. Increased consumption of these energy-dense foods alongside an increase in sedentary behaviour among children creates an imbalance (difference in calories consumed and calories utilized through activity), which further contributes to obesity among children [15,18]. To combat the childhood obesity crisis, government and non-governmental health organizations such as the World Health Organization (WHO), and Health Canada have called for legislative changes to restrict the marketing of unhealthy foods to children [19,20]. 

Although the effects of food and beverage marketing on children are known, there are no statutory regulations in Canada or the United States that limit the marketing of unhealthy foods to children. In both countries, food marketing is self-regulated by the food and beverage industry and specifically by the Canadian Children’s Food and Beverage Advertising Initiative (CAI) and the US Children’s Food and Beverage Advertising Initiative (CFBAI) [21,22]. Composed of major food and beverage companies (Coca Cola, McDonalds, Nestle, and others), these initiatives seek to limit the advertisement of unhealthy food products to children under age 12 in traditional media, new media, and in schools [21,22]. However, both initiatives have been found to be ineffective in limiting the advertisement of unhealthy food and beverages to children, as many of the products that they deem “permissible” do not meet nutritional standards [17,23,24]. Since their inception, the healthfulness of the products advertised to children by the participating companies has remained poor [17,23,24]. 

While there has been research documenting food and beverage marketing to children in traditional media, new media, and in schools [10,11,12,17,23,24], there is a lack of research assessing the presence of food and beverage marketing in other environments where children spend time, including in movie theatres. Visited by people of all ages, movie theatres are prime environments for food and beverage advertisements, as the sales of these products are an important source for revenue for movie theatres [25]. Since the 1980s, advertisers have found movie theatres to be a promising avenue for promotion due to the attention of moviegoers towards the large movie theatre screens [26]. In addition, some large movie theatres have arcades, a variety of food concession stands and a dining area akin to a food court. As such, some moviegoers may be spending extended periods of time in movie theatres, thereby increasing the opportunity for exposure to food marketing in this environment. According to Cineplex, a large movie theatre chain in Canada, advertising in their movie theatres leads to a 58% increase in advertising awareness, an 86% increase in correct brand association, and a 39% increase in product likeability [27]. Furthermore, research suggests that moviegoers are more likely to recall ads in movie theatres compared to television, print, and radio ads, with restaurant ads being recalled the most amongst moviegoers [28,29]. The impact of movie theatre advertising is concerning when one considers the high level of movie theatre attendance. According to Cineplex, 69 million visits are made to their theatres annually [30]. In Regal Cinemas (a major American chain), 170 million people attend annually [31]. Of the 1.3 billion movie tickets sold in Canada and the US in 2016, 11% were for children age 2–11 years [32]. While there have been some reports on the unhealthy nature of some foods sold in movie theatres, there has been little research on the frequency and healthfulness of foods and beverages marketed in this environment [33,34,35]. 

The objective of this pilot study was to examine children’s potential exposure to unhealthy food advertising in movie theatres in Ontario (Canada) and Virginia (US). To do so, this study quantified instances of food and beverage marketing in this environment as well as assessed the healthfulness of promoted foods and beverages. 

## 2. Methodology

### 2.1. Movie Theatre Sampling

A convenience sample of 14 large movie theatres in Ottawa, Ontario (Canada) (*n* = 7), and three different areas surrounding Richmond, Virginia (USA) (*n* = 7) were sampled. Data collection was undertaken in both Canada and the United States in order to inform whether movie theatre advertising merits further study in these countries. These two particular cities in Canada and the United States were selected for reasons of convenience, as participating researchers were either located in Ottawa or in and around Richmond, Virginia. Major corporate theatres with multiple locations were selected, as it was deemed that they would be more representative of movie theatre environments that most people patronise. 

### 2.2. Environmental Assessment

In each theatre, an environmental scan was conducted in order to assess movie theatre advertisements from multiple sources (e.g., screens, signs, banners, cardboard cut-outs, magazines), and the pre-screening advertisements that played before children’s movies. Each movie theatre was visited at least once a month for a 4-month period (February to May 2019). An environmental assessment was conducted during the first (February) and last (May) month of data collection. The assessment of advertisements on screens located in the movie theatre that were outside of the actual screening rooms was conducted at every visit. The assessment of the pre-screening ads (shown before the actual movie begins) was conducted at each movie theatre every month. A total of 28 visits were made to theatres in Ontario, while 36 were made in Virginia.

The environmental assessment involved recording the presence and number of theatre food concession stands, other food concession stands (i.e., those selling foods under a different brand than that of the movie theatre e.g., Pizza Pizza, Outtakes Bistro, Starbucks), food displays (defined as those that stand alone, not a part of the concession stands) and vending machines, on a smartphone in a pre-established template. Various types of food advertisements were also recorded and included cardboard cut-outs, advertisements on screens located outside the screening rooms, in party rooms and washrooms, and in free theatre magazines as well as other signs, stickers, and banners located inside and outside the movie theatre. Food advertisements included all ads that featured food or beverages, including both ads where food was the main product (e.g., movie theatre popcorn ad) as well as those where it was not (e.g., Interac ad featuring a bowl of candy). The promoted food and beverages within these ads were also noted. Advertisements on screens outside the theatre rooms were watched for an entire loop (until the loop repeated itself), and all food and beverage ads were recorded. 

### 2.3. Pre-Screening Advertisements

Every week, a list of G (General audience) and PG (Parental Guidance) rated movies playing in selected theatres in Ontario and Virginia was created. Each data collector (SW and FH) independently determined if the movies on the list were considered children’s movies. For the purposes of this study, “children” was defined as anyone under 13 years of age. A movie was considered a children’s movie using the following criteria: the movie had to be rated G or PG and include either themes that appeal to children (e.g., fantasy, magic, suspense, adventure, or virtual worlds); characters who were either children, child-like, or appeal to children (e.g., super heroes, talking animals, or imaginary creatures); be based on children’s stories (e.g., Winnie the Pooh) or have a story that is told from the child’s perspective. The ratings for each of the movies were obtained and confirmed from the respective movie theatre website(s). During each visit, a children’s movie from the list was randomly selected. At the next theatre, the movie that was previously seen would be excluded from the list until all children’s movies that were currently screening and on the list were seen. If all movies had been previously seen, then a movie would be seen twice, after which the movie selection process would restart as per the methodology until a new movie started screening. A list of the movies viewed in Ontario and Virginia is available in Appendix A. 

The pre-screening ads in this study were defined as advertisements that appeared on the screen in the screening room 20 minutes before the posted movie start time and until the actual start of the movie. For each food ad identified, the following data were collected: the name of the parent company and the nutritional information of promoted foods. The food ads were also classified by food category and as either product or brand ads. Product ads featured actual products, while brand ads were food ads that featured the company logo but no specific product. As for food categories, a list based on previous Canadian research [23] and tailored to the current study was used. The food categories were alcohol, candy and chocolate, cold cereal, coffee, compartment snacks, cakes and cookies, fast food restaurants, pizza, popcorn, restaurants, savoury snacks (other than popcorn), regular soft drinks, diet soft drinks, other sweetened beverages, food courier services and other. Advertisements promoting multiple foods sold by theatre concession stands were categorised into multiple food categories, while advertisements promoting foods sold by fast foods restaurants, including other food concession stands in the movie theatre, were categorised as fast food, regardless of promoted products. Food ads were also classified as to whether they promoted a food or beverage product sold in the movie theatre environment. The same information was collected for the food ads identified in movie theatre environments. These ads were also similarly classified.

### 2.4. Nutritional Analysis

The nutritional information was recorded for all advertised food and non-alcoholic beverages. This information was first sought on the Canadian company websites for products advertised in Ontario and on the US company websites for products advertised in Virginia. If the products could not be found on either website, then the Canadian Nutrient File or the company’s international website was referred to for product information. Information collected included energy, total fat, saturated fat, trans fat, sodium, carbohydrates, fiber, sugar, and protein per stated serving. In cases of ads promoting multiple product (e.g., combos), nutritional information was collected for each item and assessed individually. We assessed the nutritional content of promoted food products using the World Health Organization (WHO) Nutrient Profile Model (NPM) for Europe [36]. The WHO NPM categorizes food items into 17 distinct food categories: chocolates and confectionaries, cakes and pastries, savory snacks, beverages, edible ices, breakfast cereals, yogurt and milk products, cheese, ready-made convenient foods, butter and other fats or oils, bread, pasta and grains, fresh and frozen meat or fish, processed meat or fish, fresh and frozen fruits or vegetables, processed fruits or vegetables, and sauces. Certain food categories such as chocolate and confectionaries, cakes and cookies, edible ices, energy drinks, juices, and other beverages containing added sugar and non-sugar sweeteners are prohibited from advertisement to children, regardless of their nutritional contents while fresh or frozen meats, vegetables, fish, and fruits are always permitted for advertisement. Products in the remaining food categories are permissible for advertising to children unless they exceed established thresholds for total fat, saturated fat, total sugars, added sugars, non-sugar sweeteners, salt, and/or energy per 100 g/100 mL serving. As per the WHO NPM, advertisements featuring multiple products (e.g., combo meals) were considered not suitable for advertising, if at least one of the products advertised exceeded the established nutrient content limits. Also, 17.8% of food advertisements identified in the movie theatre environment and 11.2% of pre-screening ads could not be classified, as they exclusively promoted foods that are not included in the WHO’s NPM (e.g., coffee, alcohol, brand ads; *n* = 399) or because of missing nutrition information (*n* = 8). 

### 2.5. Statistical Analysis

Statistical analyses were conducted using SPSS statistics for Windows version 26 (IBM Corp., Armonk, NY, USA). The median number of signs/banners, advertising screens, food concession stands, food product displays, magazine advertisements, and vending machines was tabulated. The promoted food categories and companies, the presence of advertisements promoting foods sold in theaters, and the healthfulness of the food advertisements were described using frequencies and presented by location. The average number of food ads screened before children’s movies in Ontario and Virginia was also calculated. 

## 3. Results

### 3.1. Marketing in the Movie Theatre Environment

During the environmental scan in February and May, the median number of signs and banners in Ontario movie theatres was 35 (min-max: 10–87). In Virginia, the median number of signs and banners was 2 (min-max: 0–3). The median number of signs and banners promoting food in Ontario was 2 (min-max: 0–13), and in Virginia, there was none. The median number of screens advertising in Ontario movie theatres was 13 (min-max: 10–28) and 2 (min-max: 2–4) in Virginia. While there were no free magazines in any theatres in Virginia, the four theatres in Ontario distributed such magazines, and the median number of food ads in these magazines was 3 (min-max: 2–4). In Ontario, the median number of movie theatre concession stands and other types of food concessions was 1 (min-max: 1–3) and 2 (min-max: 0–4), respectively. The other food concession stands present in movie theatres in Ontario were Pizza Pizza (*n* = 1 theatre), Starbucks (*n* = 2), Outtakes Bistro (*n* = 4), Poptopia (*n* = 2), YoYo’s Frozen Yogurt (*n* = 4), and a VIP lounge (*n* = 1). The median number of movie theatre concession stands in Virginia was 3 (min-max: 1–4), but there were no other types of food concession stands present. In terms of food displays, there was a median number of 6 in Virginia (min-max: 4–8) and a median number of 2 in Ontario (min-max: 0–5). The types of food in the food displays in Ontario were candy and chocolates (*n* = 5 theatres) and frozen yogurt toppings (*n* = 1). In Virginia, only candy and chocolate were in food displays (*n* = 7 theatres). Movie theatres in Ontario had a median number of 2 (min-max: 1–4) vending machines, and these sold Coca Cola beverages and Mini Melt ice cream. No vending machines were present in movie theatres in Virginia. Two movie theatres in Ontario and six in Virginia sold food combos for children that included popcorn, soft drinks, and candy or chocolate, among other foods (e.g., nachos, hot dogs). Three movie theatres in Ontario also had arcade games where game rewards could be redeemed for candy or chocolate.

Overall, there were 1999 food advertisements identified in the movie theatre environments in Ontario and 43 identified in Virginia. The five most advertised food categories in Ontario movie theatres were popcorn (37.1% of food ads), alcohol (13.9%), candy and chocolate (13.6%), diet soft drinks (10.0%), and regular soft drinks (6.6%). The five most advertised food categories in Virginia were soft drinks (97.7% of food ads), popcorn (81.4%), candy and chocolate (51.2%), other savory snacks (27.9%), and other (16.3%) (see Table 1). In Ontario, 91.5% of food advertisements (*n* = 1829) promoted a food product sold in the movie theatre, while this was the case for 100% of advertisements (*n* = 43) in Virginia. A total of 28 different companies advertised using food and beverages in movie theatre environments in Ontario, while 3 different companies advertised in Virginia.

As shown in Table 2, most or all food ads in Ontario (99.4%) and Virginia (100%) were classified as not permissible for advertising to children.

### 3.2. Pre-Screening Ads 

Overall, 241 pre-screening advertisements promoting food/beverages were seen before the start of 28 children’s movies in Ontario compared to 80 food advertisements that were seen in Virginia before 36 children’s movies. In Ontario, the average number of food ads seen per movie was 8.6 (SD = 3.3), while in Virginia, the average was 2.2 (SD = 0.9). The five most advertised food categories in Ontario were popcorn (44.8% of food ads), candy and chocolate (13.3%), dine-in restaurants (12.4%), coffee (8.7%), and regular soft drinks (7.1%). The only three food categories advertised in Virginia were popcorn (100% of food ads), soft drinks (85.0%), and candy and chocolate (8.8%) (Table 3). In Ontario, 77.6% of food advertisements (*n* = 187) promoted a food product sold in the movie theatre, while this was the case for 100% of advertisements (*n* = 80) in Virginia.

Twenty-two different companies advertised food/beverages before the screening of children’s movies in Ontario, whereas only four different companies did so in Virginia. As shown in Table 4, the most advertised companies in Ontario were Cineplex (57.3%), Coca Cola (12.9%), Landmark (4.1%), and Mars (4.1%) while in Virginia, these were Regal (93.8%) and Coca Cola (85.0%). Most or all food ads seen before children’s movies in Ontario (97.1%) and Virginia (100%) were classified as not permissible for advertising to children (Table 2).

## 4. Discussion

This pilot study found that most food advertisements identified in selected movie theatres in Ontario and Virginia, including those advertised before children’s movies, promoted products considered not permissible for advertising to children according to the WHO’s European NPM. Overall, the most advertised food products were popcorn, soft drinks, and chocolate and candy: the classic triad of movie theatre foods. Recent research has shown that movie theatre popcorn contains anywhere between 400 and 1200 calories, while one chocolate bar or package has between 300 and 1100 calories [35]. Soft drinks, depending on size, have approximately 130–310 calories and 33–78 g of sugar [35,37]. Should one consume this triad together, one could consume between 850 and 2800 calories in one sitting of about 2 hours. In general, television-viewing environments encourage children to eat (especially when food advertisements are being aired), and watching television and eating at the same time is associated with childhood obesity [38]. Unsurprisingly, the healthfulness of advertisements noted in this study reflects the advertising of food and beverages to children on television and in digital media [17,23,24,39,40,41]. These findings are concerning, given that food and beverage advertising is known to influence children’s caloric intake following exposure [14] and children may not compensate for this increased intake at subsequent meals [15]. As such, the large volume of food and beverage advertisements in movie theatre environments along with the ease of access to most advertised products and the sedentary nature of movie viewing could potentially be contributing to obesity among children internationally. Although movie theatres might not be visited often, this environment contributes to children’s cumulative exposure to food and beverage advertising.

Although our study did not intend to compare the movie theatre environments in Ontario and Virginia, it is interesting to note that there was more advertising and a greater diversity of food categories and companies/brands promoted in movie theatres in Ontario compared to those in Virginia. This could be the result of differing marketing strategies and the fact that in Ontario, one theatre company also owns some of the restaurants (YoYos, Poptopia, and Outtakes Bistro) that they respectively advertise and house in their theatres [25]. As such, it would make sense for them to advertise their own restaurants to increase their overall revenue. This finding mirrors the increase in diversity of food items available in movie theatres, as they aim to have more restaurants and sit-in options [25]. This difference in advertising between Ontario and Virginia could also be due to differences in movie theatre sizes. Generally, movie theatres in Ontario consisted of anywhere between 10 and 15 movie-screening rooms, while in Virginia, each movie theatre had between 10 and 12 movie-screening rooms. 

### 4.1. Alcohol 

In addition to the advertising of unhealthy food and non-alcoholic beverages, alcohol was also promoted in 13.9% of food advertisements in the movie theatre environment and 2.5% of advertisements screened before children’s movies in Ontario. According to data from 2016, adolescents aged 12–17 years in the United States and Canada go to movie theatres almost twice as often as children under 11 (6.1 visits per year, on average compared to 3.3) and account for 13% of movie tickets sold, even though they constitute 8% of the population [32]. As such, adolescents are also at risk of being exposed to alcohol advertisements in this environment. The potential exposure to alcohol advertising among children and adolescents in movie theatres is concerning, given that systematic reviews of longitudinal studies suggest that exposure to alcohol marketing has a dose–response relationship with alcohol consumption and increases the likelihood of alcohol consumption at an earlier age as well as binge drinking among young people [42,43]. Given that alcohol consumption is associated with more than 30 health conditions [44], in addition to other harms [45], the WHO and other health organisations have called for more public policies aimed at reducing alcohol consumption, including greater restrictions to alcohol marketing [46,47] 

While the content of alcohol advertisements cannot target children under the drinking age, alcohol advertisements in movie theatres are permitted in Ontario but “may not run in conjunction with movies which have a “Suitable for All” (G) rating.” [48]. As for movies rated PG, provincial guidelines state that these “have to be dealt with cautiously to ensure the movie itself is not targeted specifically at persons under the legal drinking age” [48]. In our sample, six advertisements featuring alcohol, along with other foods, screened before movies rated PG including *Detective Pikachu, Missing Link, Captain Marvel, Avengers End Games*, and *Shazam!* While it could be argued that *Captain Marvel, Avengers*, and *Shazam!* do not “specifically target” those under the legal drinking age, the same cannot be said of *Missing Link* or *Detective Pikachu*, which are both animated movies, one of which is based on a video and card game popular with children. Our findings suggest that the current guidelines in Ontario do not adequately protect children from being exposed to alcohol advertisements screening before movies they are likely to see.

Notably, all alcohol advertisements in Ontario were also identified in movie theatres where alcoholic beverages were currently or soon to be available for sale. Research suggests that advertising originating from locations licensed for on-site consumption of alcohol constitute an important source of exposure to alcohol advertising among children. For instance, a study quantifying children’s exposure to alcohol marketing in New Zealand found that children aged 11 to 13 years old are exposed to an average of 4.5 alcohol promotions per day (excluding within stores, on packaging, and on screens), and 18.7% of this exposure occurred in establishments that sell alcohol for on-site consumption [49]. Our findings suggest that greater restrictions may be needed to protect children and adolescents from promotional activities related to alcohol vending in movie theatre environments.

### 4.2. Strengths and Limitations

This study examined the frequency and healthfulness of food advertising in a small convenience sample of movie theatres in Ontario and Virginia over a four-month period. Although selected movie theatres were part of large movie theatre chains and were visited monthly to account for some seasonal variations, our findings are not representative of the advertising in movie theatres over the calendar year nor of all movie theatres in either country. As such, nothing can be inferred from noted differences between movie theatres in Ontario and Virginia. There was also variation in the types of movies where the pre-screening ads were collected. The ads were collected from movies that were rated for children’s ages (G rating) but were also collected from movies that appealed to children with a PG rating (such as superhero films). Due to this difference in age rating between chosen movies, there may have been a difference in the products being advertised. Another limitation of this study is that advertisements promoting products other than food were not recorded. As a result, the frequency of food advertisements relative to other types of ads in this environment cannot be ascertained. Lastly, this study only documented advertisements in the movie theatre environment and before the screening of children’s movies and did not include the presence of product placements within movies. Notwithstanding these limitations, this is the first study to examine food and beverage marketing in movie theatre environments, including advertisements shown before children’s movies.

### 4.3. Policy Implications

The predominantly unhealthy nature of foods advertised before children’s movies highlights the potential inadequacy of current self-regulatory policies in both Canada and the United States. Currently, the Children’s Food and Beverage Advertising Initiatives of both countries do not apply to movie theatres [21,22]. Even if they did, it is doubtful that these initiatives would adequately protect children, as they do not involve all food companies, and their industry-developed nutrition criteria for defining which products can be advertised to children have been found to be lax [23,24]. As such, it is recommended that movie theatre environments be considered for inclusion in statutory food marketing restrictions in order to protect children’s health. Such restrictions have been recently adopted in Chile, where the advertising of unhealthy foods to children in movie theatres and on television is restricted from 6am to 10pm in order to reduce the prevalence of childhood obesity [50]. In 2016, Bill S-228 was introduced in Canada, which sought to prohibit unhealthy food and beverage marketing to children in a variety of media and settings; however, this bill failed to pass due to intense industry lobbying [51,52]. Despite this failure, the current government remains committed to developing food marketing restrictions [53]. Our findings suggest that settings where children gather, including movie theatre environments, should be further studied and considered for inclusion in advertising restrictions. In movie theatres, this pilot study suggests that in order to be effective, marketing restrictions should not only apply to movies that are specifically targeted to children but should also apply to those that target general audiences that are popular with children and families (e.g., superhero movies). Given the volume of unhealthy ads identified in the movie theatre environment (e.g., entrance, lobby, concession area etc.) statutory restrictions should also likely apply to the whole environment (not just “child-targeted” movies). More research on food marketing in movie theatres, including an assessment of whether and what child-appealing marketing techniques are being used, is needed to inform policy in this area.

## 5. Conclusions

This pilot study suggests that movie theatres in Ontario and Virginia may expose children to a high volume of unhealthy food and beverage advertisements in the environment and before the screening of children’s movies. To inform policies, further research is needed to determine if movie theatre settings should be included in unhealthy food and beverage marketing restrictions aimed at protecting children’s health.

## Figures and Tables

**Table 1 nutrients-12-01253-t001:** Frequency of advertisements by food category in the movie theatre environment in Ontario and Virginia from February to May 2019 ^1^.

	Ontario *N* = 1999	Virginia *N* = 43
	*n* (%) ^2^	*n* (%) ^2^
Alcohol	278 (13.9)	0 (0)
Candy and Chocolate	272 (13.6)	22 (51.2)
Cereal	20 (1.0)	0 (0)
Coffee	98 (4.9)	0 (0)
Compartment Snacks	4 (0.2)	0 (0)
Cakes and Cookies	3 (0.2)	0 (0)
Food Courier Service	69 (3.5)	0 (0)
Dine-In Restaurants	120 (6.0)	0 (0)
Fast Food Restaurants	128 (6.4)	0 (0)
Pizza	32 (1.6)	0 (0)
Popcorn	742 (37.1)	35 (81.4)
Other Savory Snacks	54 (2.7)	12 (27.9)
Soft Drinks (regular)	132 (6.6)	42 (97.7)
Soft Drinks (diet)	199 (10.0)	0 (0)
Other Sugar-Sweetened Beverages	45 (2.3)	0 (0)
Other	6 (0.3)	7 (16.3)

^1^ The advertisements within the movie theatre environment included those that were displayed on screens, which were assessed once per month from February to May 2019, as well as other types of advertisements (e.g., signs, banners, magazines ads), which were only assessed in February and May 2019. The “movie theatre environment” excludes ads screened before children’s movies. ^2^ The sum of frequencies exceeds the total number of advertisements, as some ads promoted multiple foods categories.

**Table 2 nutrients-12-01253-t002:** Healthfulness of food advertisements located in the movie theatre environment and before the screening of children’s movies in Ontario and Virginia in February–May 2019 according to the WHO European Nutrient Profile Model (NPM).

**Movie Theatre Environment Ads**		
	**Ontario ^1^** ***N* = 1636**	**Virginia** ***N* = 43**
	*n* (%)	*n* (%)
Permitted	10 (0.6)	0 (0)
Not Permitted	1626 (99.4)	43 (100)
**Pre-Screening Ads**		
	**Ontario ^1^** ***N* = 205**	**Virginia** ***N* = 80**
	*n* (%)	*n* (%)
Permitted	6 (2.9)	0 (0)
Not Permitted	199 (97.1)	80 (100)

^1^ In Ontario, the permitted advertisements in the movie theatre environment promoted fried chicken (*n* = 10), while those screening before children’s movies featured canned soup and beans (*n* = 2) and pad thai (*n* = 4). WHO, World Health Organization. Bold is the other category of data.

**Table 3 nutrients-12-01253-t003:** Frequency of advertisements by food category advertised before children’s movies in Ontario and Virginia in February–May 2019. ^1^

	Ontario *N* = 241	Virginia *N* = 80
	*n* (%) ^2^	*n* (%) ^2^
Alcohol	6 (2.5)	0 (0)
Candy and Chocolate	32 (13.3)	7 (8.8)
Cereal	4 (1.7)	0 (0)
Coffee	21 (8.7)	0 (0)
Cakes and Cookies	6 (2.5)	0 (0)
Dine-In Restaurants	30 (12.4)	0 (0)
Fast Food Restaurants	16 (6.6)	0 (0)
Food Courier Service	9 (3.7)	0 (0)
Pizza	9 (3.7)	0 (0)
Popcorn	108 (44.8)	80 (100)
Other Savory Snacks	13 (5.4)	0 (0)
Soft Drinks (regular)	17 (7.1)	68 (85.0)
Other Sugar-Sweetened Beverages	4 (1.7)	0 (0)
Soft Drinks (diet)	10 (4.1)	0 (0)
Other	3 (1.2)	0 (0)

^1^ 28 and 36 children’s movies were seen in Ontario and Virginia, respectively. ^2^ The sum of frequencies exceeds the total number of advertisements, as some ads promoted multiple foods categories.

**Table 4 nutrients-12-01253-t004:** Frequency of advertisements by food company advertised before children’s movies in Ontario and Virginia in February–May 2019. ^1^

	Ontario *N* = 241	Virginia *N* = 80
	*n* (%) ^2^	*n* (%) ^2^
Bowtie Movies	-	5 (6.3)
Campbell Company	1 (0.4)	-
Cineplex	138 (57.3)	-
Coca Cola	31 (12.9)	68 (85.0)
Dairy Queen	3 (1.2)	-
Dr Oetker	4 (1.7)	-
General Mills	6 (2.5)	-
Go4Greek	4 (1.7)	-
Hershey	4 (1.7)	-
Landmark	10 (4.1)	-
Loblaws	1 (0.4)	-
Mars	10 (4.1)	7 (8.8)
McDonalds	4 (1.7)	-
Mondelez International	4 (1.7)	-
Nestle	2 (0.8)	-
Non-food company	37 (3.6)	-
Pizza Pizza	4 (1.7)	-
Regal	-	75 (93.8)
Restaurant Brand International	1 (0.4)	-
Skip the dishes	4 (1.7)	-
UberEats	5 (2.1)	-
The Wendy’s Company	4 (1.7)	-

^1^ 28 and 36 children’s movies were seen in Ontario and Virginia, respectively. ^2^ The sum of frequencies exceeds the total number advertisements as some ads promoted multiple companies.

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
