# Peer review of "The Frequency and Healthfulness of Food and Beverage Advertising in Movie Theatres: A Pilot Study Conducted in the United States and Canada"

_nutrients, 2020, doi:10.3390/nu12051253_

Round 1
Reviewer 1 Report
After the second round of review, I don't think the authors have adequately addressed my previous reviews. The questions remain 1. why the case study of both US and Canada are presented, without a clear discussion of the design. From the response letter, it reads like the data was collected due to the availability of research assistants in these sites. If so, this shows a lack of design. It may be better to only present the US data. 2. The link between the results and policy implications is still weak. I understand this is a retrospective data, and that health outcomes were not included in the original surveys. However, the authors could use existing studies to indicate potential impacts. At least, a clear discussion between ads viewing and the nutritional outcome should be included given the journal's scope. The current discussion focuses on food marketing.
Author Response
Reviewer 1
Comments and Suggestions for Authors
After the second round of review, I don't think the authors have adequately addressed my previous reviews. The questions remain 1. why the case study of both US and Canada are presented, without a clear discussion of the design. From the response letter, it reads like the data was collected due to the availability of research assistants in these sites. If so, this shows a lack of design. It may be better to only present the US data. 2. The link between the results and policy implications is still weak. I understand this is a retrospective data, and that health outcomes were not included in the original surveys. However, the authors could use existing studies to indicate potential impacts. At least, a clear discussion between ads viewing and the nutritional outcome should be included given the journal's scope. The current discussion focuses on food marketing.
- We now justify collecting data in both Canada and the United States.
Lines 94-99 now read:
“A convenience sample of fourteen large movie theatres in Ottawa, Ontario (Canada) (n = 7), and three different areas surrounding Richmond, Virginia (USA) (n = 7) were sampled. Data collection was undertaken in both Canada and the United States in order to inform whether movie theatre advertising is present and merits further study in these countries. These two particular cities in Canada and the United States were selected for reasons of convenience, as participating researchers were either located in Ottawa or in and around Richmond, Virginia.
- The link between our results and the policy implications have been made as clear as possible given that this is a pilot study. In particular, in Lines 273-274 of the Discussion we say:
“In general, television-viewing environments encourage children to eat (especially when food advertisements are being aired) and watching television and eating at the same time is associated with childhood obesity [35]. Unsurprisingly, the healthfulness of advertisements noted in this study reflects the advertising of food and beverages to children on television and in digital media [36, 37, 38]. These findings are concerning given that food and beverage advertising is known to influence children’s caloric intake following exposure [14] and children may not compensate for this increased intake at subsequent meals [15]. As such, the large volume of food and beverage advertisements in movie theatre environments along with the ease of access to most advertised products and the sedentary nature of movie viewing could potentially be contributing to obesity among children internationally. Though movie theatres might not be visited often, this environment contributes to children’s cumulative exposure to food and beverage advertising.”
Potential impacts of food marketing are also clearly stated in the Introduction where we indicate in Lines 42- 45, the impact of food marketing on children:
“One factor contributing to obesity and poor dietary behaviours amongst children is the pervasive presence of unhealthy food and beverage marketing in traditional media (television and print media), new media (digital and social media), and in schools [8 – 12].”
We also specifically mention research on the impact of movie theatre advertising in lines 75-79:
“According to Cineplex, a large movie theatre chain in Canada, advertising in their movie theatres leads to a 58% increase in advertising awareness, an 86% increase in correct brand association, and a 39% increase in product likeability [25]. Furthermore, research suggests that moviegoers are more likely to recall ads in movie theatres compared to television, print, and radio ads, with restaurant ads being recalled the most amongst moviegoers [26, 27].”
Reviewer 2 Report
Thanks for inviting this paper. It addresses an interesting and critical issue regarding the food and beverage advertisement exposure in the children’s experience in a particular setting, i.e. movie theaters. Generally, this paper is well laid out, although the data being more descriptive, and provides some insights into advertising restrictions related to obesogenic products among growing children. The concept or research scheme is well framed around the food and beverage ads shown in movie theaters. However, the only thing I am a bit confused about is the definition of accounts or numbers of ads. Take Table 1 as an example, I could not figure out how to count dine-in restaurant and alcohol on the same measurement scale, let alone comparing the difference between Ontario and Virginia. Meanwhile, was there any inter-rater difference when researchers counted the ads? Except these mentioned methodological concerns, this paper has given a clear message on this concerning issue of unhealthy food and beverage ads in movie theaters that might be overlooked by current regulations.
Author Response
Reviewer 2
Comments and Suggestions for Authors
Thanks for inviting this paper. It addresses an interesting and critical issue regarding the food and beverage advertisement exposure in the children’s experience in a particular setting, i.e. movie theaters. Generally, this paper is well laid out, although the data being more descriptive, and provides some insights into advertising restrictions related to obesogenic products among growing children. The concept or research scheme is well framed around the food and beverage ads shown in movie theaters. However, the only thing I am a bit confused about is the definition of accounts or numbers of ads. Take Table 1 as an example, I could not figure out how to count dine-in restaurant and alcohol on the same measurement scale, let alone comparing the difference between Ontario and Virginia. Meanwhile, was there any inter-rater difference when researchers counted the ads? Except these mentioned methodological concerns, this paper has given a clear message on this concerning issue of unhealthy food and beverage ads in movie theaters that might be overlooked by current regulations.
- The title of Tables 1, 3 and 4 has been re-named to avoid confusion. Each frequency count represents 1 advertisement.
- Inter-rater reliability was not calculated as only one rater counted the instances of advertising in each of Ottawa and Richmond during the data collection phase. Two raters were not utilized in this situation as error in counting ads is typically minimal and ad counts are objective. The number of ads were further verified by having the raters watch through the entire ad loop more than once to ensure that the ads were in fact running in a loop and that the number of ads shown were correct.
This manuscript is a resubmission of an earlier submission. The following is a list of the peer review reports and author responses from that submission.
Round 1
Reviewer 1 Report
GENERAL
This paper is interesting. It represents novel work. The research is both important and relevant. Unfortunately, the methods do not seem not sound. A bottom line comes from the authors own words:
Lines 323-324: “findings are not representative … As such, nothing can be inferred”
Despite the admission above, conclusions over-reach:
Lines 360-361: “Therefore, the development of legislation to regulate the advertising of unhealthy foods and beverages to children ages 2 to 16 should include advertising in movie theatre”
I think this study might be better cast as pilot work. The paper (which is very dense) might be revised to a research brief. I make both more specific observations and recommendations below.
SPECIFIC
Abstract:
- The abstract suggests comparability between the n=1999 ads in Ontario and n=43 in Virginia. These quantities are not comparable; they are based on both different numbers of visits and different protocols (more on that later)
Introduction:
- Line 34: “obesity” defined how? Same definition used by the three sources cited—WHO, US, Canada?
- Line 47; “caloric gap” suggests a deficit, not an excess. Maybe “imbalance” or “mismatch” or even “excess” instead.
- Lines 57, 59, and 61: “initiatives”, “both initiatives”, and “their” inception? These plural constructions all seem to be referring to the singular “Children’s Food and Beverage Initiative.” What is the other initiative?
- Lines 64-67: children spend hours and hours—daily—with traditional media, with new media, and in schools. How often do kids go to the movies? Maybe a few times per year for a couple hours at a time? The authors need to convince the reader why theatre advertisements (representing a comparatively negligible amount of experience) are an important “exposure”.
- Lines 67-69: The suggestion is that all of the products advertised are available for purchase from theatres. Table 1 demonstrates this is not true (e.g., restaurant advertising).
- Line 70: “undivided attention” to “theater screens” does not speak to the non-screen ads—the *vast* majority of ads in Ontario; a high proportion of the ads in Virginia. Also, maybe attentions was “undivided” in the 1980s (Line 69); but now kids come with their own tablets, phones, video games, etc. Additionally, some families may deliberately arrive late to *not* have to sit through the ads.
- Line 73-75: missing from the assertion that “moviegoers are more likely to recall ads” is the fact that this finding comes from only a single study—from Hong Kong. The statement (and others in the paragraph) need to be less definitive. There needs to be appropriate recognition of the limitations of cited research.
Methodology:
- An overarching comment: a logical unit for comparison would be “a theatre visit.” In other words, how many ads might a given child encounter on a trip to the theater. Authors could report a range based on what movie was actually seen. I can think of at least 5 useful stratifications: (1) Ontario vs. Virginia, (2) screen ads vs. ads elsewhere in the theatre, (3) “pure ads” vs. product displays also serving as ads, (4) PG vs. G-rated movies, (5) washroom the child might use (if there were differences in advertising for men’s vs. women’s vs. gender neutral washrooms)
- As it stand now, there are different numbers of visits (28 in Ontario, 36 in Virginia) as well as different procedures used at each visit (sometimes assessing environmental ads, sometimes not). In this context, total ads (across visits) does not seem to make much sense.
- Lines 102-104: this text makes it seem like food-product displays counted as ads (which they absolutely should) but analysis by products vs. “pure ads” is not further discussed
- Lines 108-109: so “food ads” did not include co-marketing or ‘stealth advertising’ (e.g., food-product placement—perhaps even branded—in ads for other products/services)? This is a limitation. Also, it should be clarified that “food” includes beverages. Right? Additionally, this study was only looking a “food ads”? Not *all* ads? Wouldn’t the proportion of food-to-total ads be important? E.g., Maybe 43 ads in Virginia is a lot. But if there are 2,000 ads total (there seemed to be at least that many in Ontario), does 2% of total advertisements matter?
- Lines 122-123: “At the next theatre, the movie that was previously seen would be excluded from the list”? Is there certainty that pre-movie ads were the same theater-by-theater for a given film? If not, another limitation.
- Lines 128-129: “pre-screening ads” had to occur within “20 min before posted movie start time”? Why not *all* pre-screening ads (which in U.S. theaters can start 30 min or more before the feature film)?
- Line 145-155: This whole section on “marketing techniques” is all *highly* subjective. It is not even clear how “children” are defined? Ages 2-19 (Lines 35) ? Ages 6-17 (line 36)? Age 2-16 (Line 361)? Regardless, the potential age spread is *enormous.* What appeals to a teenager relates not at all to what appeals to a toddler. “Fantasy, magic, … adventure”? Like “Game of Thrones”? (which is decidedly not child-appropriate). An appendix should show the tool used. The text should discuss inter-rater reliability. Otherwise, none of this is valid. In fact, the whole section should probably be cut.
- Lines 155-157: Different methods in the two cities? Another problem.
- Line 183: See first bullet above (overarching comment) with ideas for analyses. Additionally, some thought might be given to size of the theater—either by square footage or number of screens. This idea again gets to the point about absolute number vs. proportion of total for ad “exposure.”
Results:
- Lines 195-196: “The median number of screens advertising in Ontario movie theatres” ranged to 28?? There is a theatre with 28 screens in Ontario?
- Line 199: “concessions” meaning “concession stands” (separate entities within theatres selling food/drink items) not “food/drink products” (items sold at concession stands), right?
- Lines 204-207: “food displays” in neither country ever included popcorn??
- Lines 215-216: are “diet” soft drinks “permitted” items for advertising?
- Table 1: Title should clarify that “theatre environment” excludes ads on screens
- Table 2: what are “Other” ads? Are they for “Uber eats” or “Loblaws” (from Table 5)? If so, the numbers don’t match up. Needs explanation and reconciliation.
- Table 3: This is mostly subjective (e.g., how do you define “bright” colors? what is a “cartoon” font?) Presumably authors took photos of all ads so could go back to them with a more rigorous process. Needed are the qualitative methods for arriving at themes. Also needing detail: inter-rater reliability or the consensus process for categorizing. Actually, all of such details could be a whole separate paper. As mentioned earlier, I might cut all of this content about “marketing technique” from the current paper (and consider making the current paper a shorter research brief).
- Table 4: what were the “permitted” items advertised? Footnote please.
- Line 258: is “Table 2” supposed to be “Table 4” here?
- Line 261: is “26” a typo for “36”?
- Line 265: reference is made to “Table 3;” not clear why there is discussion of tables out of order.
Discussion:
- Line 271: “classic triad”? Classic according to whom? Is there a reference for this?
- Line 288: “ease of access to advertised products” seems to ignore all the advertising for restaurants and other businesses off-site.
- Line 294: If it is important that one theater company owns the three restaurants listed, why do *none* of these restaurants appear in Table 5?
Reviewer 2 Report
This article studies an interesting research question: advertisements towards children in movie theaters. They used a survey method from Feb to May in two cities. The conclusion should be interesting to policymakers and some non-profit organizations. I have some suggestions and hopefully will help clarify and improve this manuscript.
- As mentioned by the authors, Ottawa and Richmond are not direct comparisons. It then needs to be clarified why these two cites are selected. It seems only two categories of food ads were shown in Virginia. Is this a regulation issue in the US? Sample sizes from these two cites also vary a lot. These need to be discussed in terms of what policy implications can be derived.
- I understand the main goal is to study ads viewing and it may not be possible to have a causal outcome, but it will be great to link viewing to behaviors somehow. For example, does viewing of popcorn increase sales of popcorn sales at the theater that day? Again, I understand this may not be possible since the study has already completed. However, discussion on this with historical data will help draw out the policy implications.
- The authors should discuss why the mentioned nutrition quality measurement was employed over other possible measurements, e.g. a simple calorie count, Fulgoni, Keast and Drewnowski’s “Nutrient Rich Foods” score, etc.